# The Effect of High and Low Ambient Temperature on Infant Health: A Systematic Review

**DOI:** 10.3390/ijerph19159109

**Published:** 2022-07-26

**Authors:** Darshnika Pemi Lakhoo, Helen Abigail Blake, Matthew Francis Chersich, Britt Nakstad, Sari Kovats

**Affiliations:** 1Wits Reproductive Health and HIV Institute, Faculty of Health Sciences, University of Witwatersrand, Johannesburg 2001, South Africa; mchersich@wrhi.ac.za; 2Department of Health Services Research and Policy, London School of Hygiene and Tropical Medicine, London WC1E 7HT, UK; helen.blake@lshtm.ac.uk; 3Clinical Effectiveness Unit, Royal College of Surgeons of England, London WC2A 3PA, UK; 4Division Paediatric Adolescent Medicine, Institute of Clinical Medicine, University of Oslo, NO-0316 Oslo, Norway; britt.nakstad@medisin.uio.no; 5Department of Paediatrics and Adolescent Health, University of Botswana, Gaborone 4775, Botswana; 6Centre for Climate Change and Planetary Health, London School of Hygiene and Tropical Medicine, London WC1E 7HT, UK; sari.kovats@lshtm.ac.uk

**Keywords:** heat exposure, cold exposure, ambient temperature, SIDS, mortality, neonatal health, infant health

## Abstract

Children, and particularly infants, have physiological, anatomic, and social factors that increase vulnerability to temperature extremes. We performed a systematic review to explore the association between acute adverse infant outcomes (children 0–1 years) and exposure to high and low ambient temperatures. MEDLINE (Pubmed), Embase, CINAHL Plus, and Global Health were searched alongside the reference lists of key papers. We included published journal papers in English that assessed adverse infant outcomes related to short-term weather-related temperature exposure. Twenty-six studies met our inclusion criteria. Outcomes assessed included: infant mortality (*n* = 9), sudden infant death syndrome (*n* = 5), hospital visits or admissions (*n* = 5), infectious disease outcomes (*n* = 5), and neonatal conditions such as jaundice (*n* = 2). Higher temperatures were associated with increased risk of acute infant mortality, hospital admissions, and hand, foot, and mouth disease. Several studies identified low temperature impacts on infant mortality and episodes of respiratory disease. Findings on temperature risks for sudden infant death syndrome were inconsistent. Only five studies were conducted in low- or middle-income countries, and evidence on subpopulations and temperature-sensitive infectious diseases was limited. Public health measures are required to reduce the impacts of heat and cold on infant health.

## 1. Introduction

Climate change is one of the greatest global threats to public health, and indeed an existential threat for humankind and the natural world [1]. In all future temperature predictions, the global temperature is expected to rise [2]. Heatwaves are projected to occur more often and last longer due to climate change and ongoing urbanisation [2]. The frequency of cold days is projected to reduce, but cold spells will continue to occur [2]. High and low temperature exposure as a risk factor for morbidity and mortality is being increasingly established in epidemiological studies [3,4,5,6]. Older persons, people with comorbidities, pregnant women, and children have been identified as at risk from high temperatures [3,4,7,8]. Additionally, low socioeconomic and ethnic minority groups are at a higher risk of temperature-related adverse health outcomes [3,5,6,7,9]. Socioeconomic status may impact on factors such as access to air-conditioning and to healthcare, which may differ across countries, regions and urban versus rural populations [3,5,7,9]. Moreover, populations from a lower socioeconomic backgrounds, and urban areas are more likely to reside in warmer neighbourhoods due to high population density, sparse vegetation, and lack of green spaces, and are less likely to be able to adapt to temperature extremes [3].

Three systematic reviews in the early 2010s that assessed impacts of high temperatures on child health found some evidence that young children, especially infants under the age of 1, are vulnerable to heat. Evidence, however, was limited to very few studies [10,11,12]. There is also evidence that high temperatures cause acute increases in gastrointestinal diseases, malaria, heatstroke, renal disease, asthma, and unintentional injuries in children [10,11,12,13,14]. Studies, however, investigated children as a single age group and did not stratify by smaller age bands. Children have significantly different developmental stages between 0 and 18 years of age, in terms of metabolism, thermoregulation, and behaviours that affect their health [10,11,15,16].

Infants (children aged 0–1) may be more vulnerable to temperature extremes due to physiological, anatomical, and behavioural factors (Figure 1) [15,16,17,18,19,20]. Thermoregulation, the control of internal body temperature, is influenced by many factors, some of which include body surface area to volume ratio, sweating rate, heart and blood vessel responses, and hydration status [16]. Several factors complicate thermoregulation in infants. Infants have immature thermoregulatory mechanisms, a higher metabolic rate (more energy expenditure), and a higher sweating threshold [16,19,20,21]. Additionally, infants have a smaller blood volume and a higher heart rate, which will affect the response of the heart and blood vessels to extreme temperatures [16,19,20,21]. Anatomically, infants, and especially preterm neonates, have a higher body surface to volume ratio which increases exposure to temperature [16]. In addition to these physiological and anatomical factors, infants are particularly vulnerable, as they rely on a caregiver to safeguard them, which includes moving to a cooler or warmer environment if required, dressing appropriately, and keeping hydrated [16,20].

In this study, we aimed to explore the association between exposure to high and low ambient temperatures and acute adverse outcomes in infants. We examined the strength for evidence for these associations, and the potential causal pathways. We identify evidence on various adverse outcomes and across settings and socioeconomic circumstances.

## 2. Methods

A systematic literature review was undertaken to identify epidemiological studies that assessed the impact of short-term exposure to high and/or low ambient temperatures on infants aged 0–1 years old. Medline (Pubmed), Embase, Global Health, and CINAHL Plus were searched in July 2020 (Appendix A). Reference lists of included articles were screened for eligibility. Table 1 describes the inclusion and exclusion criteria using the PICOS framework [22]. Searches were restricted to English language and publication between 2000 and 2020.

The earliest year of publication (2000) was chosen for the relevance of the impacts of temperature on current populations. There have been significant adaptations over time to reduce the impact of temperature on adverse outcomes, such as improved housing, infrastructure, and technology [23]. Studies that aggregated temperature over longer periods such as a month are prone to bias from seasonality and were excluded as they do not meet the objectives of studying short-term temperature exposure. Lastly, diurnal temperature range and mean day-to-day temperature as exposures were excluded to meet the objectives of assessing the impacts of high and low ambient temperatures, rather than temperature fluctuations

The screening of titles, abstracts, and full texts were performed by a single reviewer (DPL). Data extraction was completed using Microsoft Excel. The Köppen–Geiger climate classification was supplemented, where studies did not have a local climate description [24]. An adapted quality assessment tool was developed to assess the quality of the studies using the Office of Health Assessment and Translation (OHAT) tool [25], which evaluates the risk of bias. Two further questions on analysis and conflicts of interest were added to the tool based on a systematic review that recommended the inclusion of nine domains for observational research: selection, exposure, outcome assessment, confounding, loss to follow-up, analysis, selective reporting, conflicts of interest, and other [26]. OHAT does not specifically consider environmental health study designs such as time series and case-crossover, hence we adapted the tool to tailor it to our research question (See Appendix A). The risk of bias in each domain was scored as: (a) definitely low, (b) probably low, (c) probably high or insufficient information, and (d) definitely high [25]. The final grading was given dependent on the individual domain grades. We considered an individual study to have a definitely low risk or probably low risk of bias if all, or most domains were rated as definitely low risk. Similarly, probably high risk and definitely high risk were rated based on the number of domains that scored probably or definitely high risk. Due to the limited number of studies, the risk-of-bias assessment results were not used to exclude studies from the analysis. Following data extraction and critical appraisal, a narrative synthesis was performed. The considerable methodological diversity and statistical heterogeneity of study findings precluded meta-analysis.

## 3. Results

The study selection process resulted in 26 studies that were included in the review (Figure 2) [27].

The studies cover a limited range of geographical locations and climate zones, with most studies from North America and Asia; only five studies were from low- and middle-income countries (LMICs) [19,28,29,30,31]. Most studies are conducted in urban populations, with five from rural areas, one of which was a combined urban-rural study [28,31,32,33,34]. Nine studies assessed the impact of temperature on all-cause infant mortality and five further studies assessed sudden infant death syndrome (SIDS) (Table 2). There were five studies each on hospital visits or admissions, and infectious diseases, and two studies reported on other conditions in the neonatal period (Table 3).

### 3.1. Temperature and Infant Mortality

There was considerable evidence that high temperatures affected infant mortality; this was shown in several studies in different regions and climate zones (USA, Spain, and South Korea) [15,21,34,35,36,37]. Heatwaves or very hot days (>95th percentile of mean temperature) were associated with increased mortality in infants in California and male infants in France [34,37]. Higher temperatures were not found to be associated with increased infant mortality in Madrid; however, this study had a higher risk of bias as compared to others that examined this outcome.

There was also evidence that low temperatures increase the risk of infant mortality [32,33,38]. Infant mortality rose when maximum temperatures declined below 6 °C in Madrid. Further, neonatal mortality increased when the temperatures were low on the day of birth, specifically in indigenous populations, and rural labourers in Sweden and Italy [32,33].

Apart from SIDS, which we consider separately below, cause-specific mortality was assessed in only two studies, one of which reported an association between very hot days and mortality relating to disorders that originate in the perinatal period in California (22nd week of gestation to first 7 days of life), while no association was detected between the specific causes of mortality and temperature in a study in Spain [21,37].

The only cause of infant death that has been more comprehensively studied in the included literature is SIDS, defined as the sudden death of an infant that remains unexplained even after a case investigation, and usually occurs during sleep [39]. There were inconsistent results for the impact of daily temperature on SIDS. In studies with a lower risk of bias, SIDS cases increased at higher temperature in USA, Canada, and South Korea [36,39,40], but no association was found in another study in USA and in Austria [21,41]. There were two studies with a high risk of bias, one conducted in the USA that found no association and another one in Taiwan that found a protective effect at higher temperatures [42,43].

### 3.2. Temperature and Infant Morbidity

Higher temperatures were associated with increased hospital visits or admissions for all-cause morbidity in infants and for heat-related morbidity in neonates across many settings (New York, rural Bangladesh, Brisbane, and Ahmedabad) [13,19,28,44]. A 0.6% (95%CI 0.1–1.1%) increased risk per interquartile range increase in temperature was reported in New York and 1.05% (95%CI 1.01, 1.10%) for hot days over 97th percentile was reported in Brisbane [13,44]. Low ambient temperature was associated with increased hospital admissions for respiratory disease in Auckland, however, the evidence is less robust as this study had a high risk of bias [45].

Evidence for the impact of temperature on infectious disease in infants was less rigorous than for other outcome categories, with three out of five studies having a definitely high risk of bias. The three studies with a high risk of bias consistently found an association between low temperature and bronchiolitis [46,47,48]. Two studies, with lower risk of bias, found evidence that Hand, Foot and Mouth Disease (HFMD) incidence in China was higher at lag 0, lag 14, and cumulatively over 14 days with as much as a 2-fold increase in incidence when temperatures were above the 95th percentile [29,30].

### 3.3. Temperature and Other Neonatal Conditions

Among newborn infants in Japan, outdoor temperature was correlated with time taken for blood to clot, as measured by the International Normalised Ratio (INR) [49]. The clinical relevance of the difference in INR is uncertain [49]. Additionally, the risk of being referred for neonatal jaundice increased with higher temperatures on the day of delivery in Nepal [31].

### 3.4. Evidence of Effect Modification by Socioeconomic Factors, Race, Rural Versus Urban Areas and Time Periods

There is inconsistent evidence of effect modification in the association between temperature and adverse outcomes by socioeconomic factors and race. Jhun et al. [39]) and Basu et al. [21] found that the effects of high temperatures on risk of SIDS and all-cause mortality were higher in black infants as compared to white infants. Karlsson et al. and Scalone et al. found an increased risk of neonatal mortality in indigenous populations [32,33]. However, three other studies found no evidence for effect modification by socioeconomic factors and race [15,36,44]. There are five studies conducted in rural areas, and all five had a positive association between temperature and adverse infant outcomes [28,31,32,33,34]. Only one study described effect modification by urban environment; a greater percentage increase in infant mortality was evident in Paris during a heatwave as compared to the rest of France [34]. There is no evidence of a difference in magnitude of temperature-infant health outcomes over time, but this is an important issue to consider.

### 3.5. Quality of Studies

The studies that assessed mortality as an outcome had overall better internal validity compared with the studies that assessed hospital admissions and infectious diseases (Figure 3). This may be due to the study designs such as time-series and case crossover that were used to assess mortality and temperature, which generally scored better for assessment of confounding and analysis as compared to traditional study designs. Many studies had a low score for confounding bias, as the analysis frequently did not take long-term time trends, seasonality and air pollution into consideration. Some studies had no conflict-of-interest disclosures and consequently scored poorly on this question. In addition, studies had exposure detection bias based on misclassification of temperature. Common reasons for misclassification of temperature were large distances from weather stations to the individuals in cohorts, or a lack of detailed information provided about temperature measures.

### 3.6. Key Findings

There was consistency across studies that higher ambient temperatures and heatwaves were associated with an increased risk of all-cause infant mortality, all-cause hospital admissions, and HFMD incidence. Mortality and hospital admissions are major health outcomes. Similarly, while HFMD is a self-limiting acute viral illness, it can sometimes be fatal and can cause serious complications such as meningitis or myocarditis (inflammation of lining of brain and heart, respectively). Further, HFMD is prevalent in Asia and is increasing in some countries with epidemic and pandemic potential [51,52]. The evidence for high temperature effects on these outcomes were robust as these studies had a low risk of bias due to better exposure assessment and/or statistical analyses that adjusted for confounders. These findings are in keeping with two previous systematic reviews in children that identified infants as a subpopulation that are at a high risk of adverse outcomes associated with exposure to higher ambient temperature and heatwaves [10,11].

Only three studies assessed the impact of ambient low temperatures on infant mortality, despite the risk of hypothermia in newborns [53]. Two of the three studies found consistent evidence that low ambient temperatures on the day of birth increased the risk of neonatal mortality, while the third found that infant mortality increased when outdoor temperatures dropped below a 6 °C threshold. Two of these studies gathered data from historical populations in the 1800′s. These data can provide context to physiology and risk to temperature exposure; however, using historical data may have limitations with respect to quantifying risk as there have since been adaptations, such as better housing and insulation, which will likely reduce exposures to cold temperature extremes. Lastly, there is some evidence that low temperatures are associated with bronchiolitis in infants.

Evidence of the relationship between temperature and SIDS is difficult to interpret. It is postulated that SIDS may be related to thermal stress in the infant, supported by evidence implicating heat-related factors such as prone sleeping, overwrapping infants, and elevated indoor room temperatures [39,41]. These factors may occur due to high ambient temperatures or, paradoxically, when cooler temperatures prompt a disproportionate warming of the infant [54]. The seemingly contradictory evidence in this review may be due to several factors. Firstly, SIDS is prone to detection bias; the thoroughness of investigations, suspicion for diagnosis and inclusion of post-mortems vary across settings. Secondly, the studies are heterogenous in design, exposure measurements and risk of bias. Thirdly, SIDS is a rare occurrence, and the number of cases may be insufficient to explore statistically. Finally, the temperature-SIDS association may be modified by other risk factors, for instance, overwrapping and bundling, or air-conditioning use that may vary across populations.

No studies were found on the short-term temperature impacts of infectious diseases such as malaria, dengue, and infectious gastroenteritis. These infections carry a high burden of disease in infants and are known to be associated with temperature [13,55]. Several reasons are hypothesised for the lack of evidence in infectious disease. Firstly, modelling studies and studies that use a wider age group were excluded in the study protocol [52,56,57]. Secondly, there is a lack of epidemiological evidence and good quality records and data-keeping from LMICs, where the burden of infectious diseases is situated. Climate change will increase transmission of many infectious diseases and hence additional evidence on these outcomes is a key research priority [13].

There is limited evidence of effect modification by socioeconomic factors and race in health–temperature associations in infants. Similarly, there is limited evidence of effect modification between urban and rural areas, but we do find evidence of temperature impacts in both urban and rural environments. The limited evidence of effect modification could be due to insufficient power for sub-group analysis, and selection bias, for example, in studies that assess only hospital records.

Only 5 of the 21 included studies were from LMICs, and entire regions such as Africa, the Middle East, and South and Central America are not represented. All five studies found a positive association between temperature and adverse infant outcomes; however, comparisons to high-income countries are limited due to methodological heterogeneity. However, there is evidence, not included in this review, that the impact of climate change on children is not evenly distributed, but occurs in LMICs that are already experiencing a higher burden of disease [21]. A study demonstrated that in Africa, climate change has already doubled the heat-related child mortality compared to what would have been expected without climate change [58]. There is an obvious mismatch between vulnerability to temperature effects and research effort, which needs to be remediated in future work.

The published literature in this review is heterogenous in exposure and outcome assessments, making it difficult to directly compare results across populations. A standardisation of some measures, for example, changes in outcome per degree Celsius change in temperature, may reduce methodological diversity.

## 4. Limitations of This Review

Only English language studies were included which may limit geographical representation. Numerous studies that assess infants within wider age groups, for example, 0–2 and 0–5 years old, have been excluded. While there is significant heterogeneity in metabolism, temperature regulation, and behaviours in children in varying age groups, the inclusion of a wider age group, for example, children 0–2 years, may provide valuable information [7,11,14,16]. Some studies were excluded due to use of temperature metrics such as diurnal temperature variation. Future research could encompass a range of temperature indices to quantify the temperature–health association more broadly. Additionally, studies that were published prior to 2000, and subsequent to the search in July 2020, were excluded which may provide additional evidence, especially for older studies on the effects of low temperatures on infant health and SIDS. Furthermore, grey literature was excluded which may result in publication bias. Lastly, the use of a single reviewer for study selection may have introduced errors during screening and data extraction.

## 5. Conclusions

The evidence collected in this review indicates that there are important, sizable, increased risk of mortality and a range of morbidities in infants exposed to high temperatures. In addition, there is some evidence of an increased risk of infant mortality and bronchiolitis with lower temperature exposures. There is, however, limited evidence from LMICs, too few studies that assess subpopulations, and no literature was identified on temperature-sensitive infectious diseases such as dengue and malaria in infants.

Global temperatures and frequency of heat waves are increasing rapidly. In 2020, there were 626 million more person-days of heatwave exposure affecting infants compared to the annual average for the years 1986–2005 [55]. The risks of outcomes related to higher temperatures will increase, but there may be a decrease in cold-related outcomes due to a decrease in the frequency and intensity of cold extremes. Future research should focus on heat impacts; however, it is still important to quantify the impacts of low temperatures due to ongoing cold spells and climate variability. Lastly, this work can be utilised, together with the growing body of research on climate change and health, to motivate for more rigorous climate mitigation and adaptation strategies.

## Figures and Tables

**Figure 1 ijerph-19-09109-f001:**
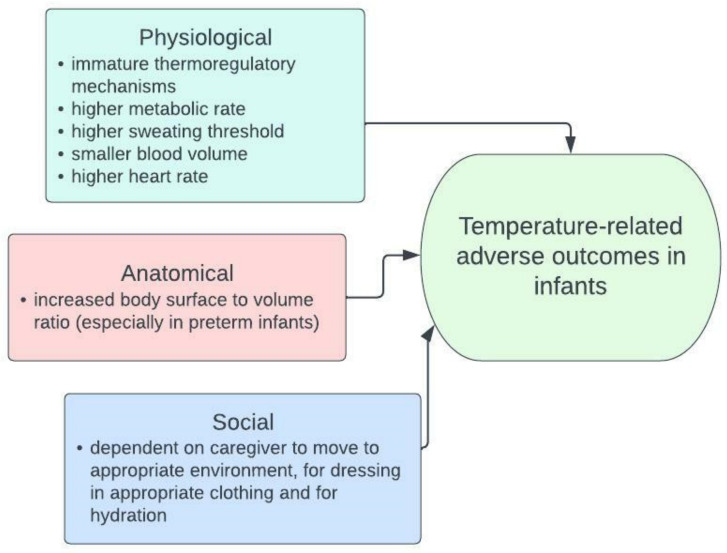
The factors that could increase temperature vulnerability in infants compared to older children and adults [15,16,17,18,19,20].

**Figure 2 ijerph-19-09109-f002:**
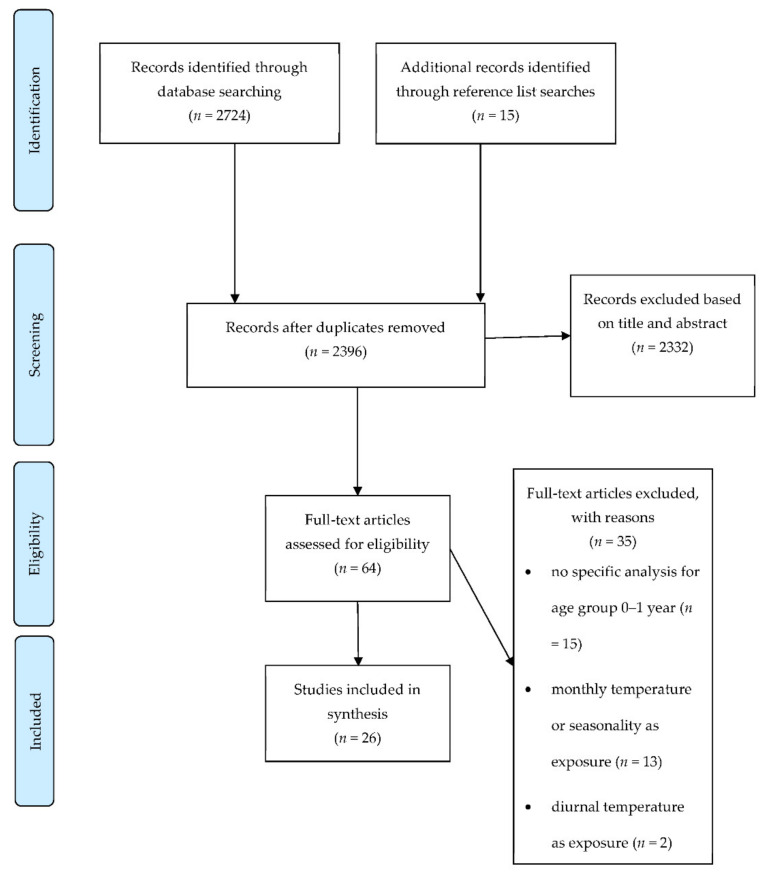
PRISMA flow diagram that illustrates the study selection process.

**Figure 3 ijerph-19-09109-f003:**
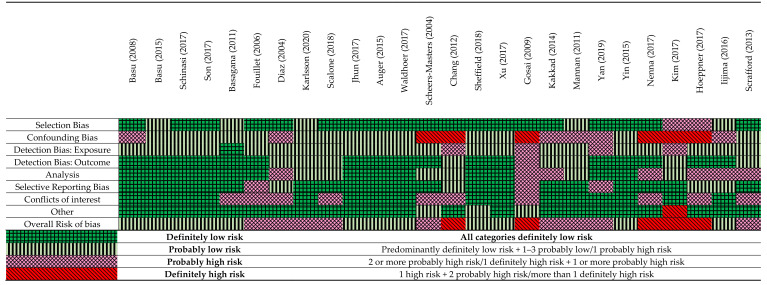
A summary of the critical appraisal using a modified OHAT Risk of bias tool [15,17,19,21,28,29,30,31,32,33,34,35,36,37,38,39,40,41,42,44,45,46,47,48,49,50].

**Table 1 ijerph-19-09109-t001:** Inclusion and exclusion criteria using the PECOS framework.

PICOS Framework	Inclusion	Exclusion
People	Results report for infants 0–1 year age group	If infants were included in wider range of age, e.g., 0–2 or 0–5 years old, and not specifically for 0–1 years
Exposure and comparator (considered together)	Weather related, high and low temperatures	Non-weather-related temperature exposures, e.g., incubators, baths, or infant body temperature
	Short term exposure, i.e., temperatures 0–14 days before event	Seasonality, monthly temperature and/or temperature exposure >14 days prior to outcome measures
	Indoor and outdoor ambient temperature	Factors such as ozone and pollution that may be secondarily affected by temperature
	Exposure after birth	Diurnal temperature range, or mean day-to-day temperature changes
		Humidity only as exposure
Outcomes	Any adverse health outcomes	Birth-related outcomes—pre-term birth, stillbirth, low birth weight, and birth defects
Setting	All countries/regions	
Type of study design	Quantitative observational study designs	Conference abstracts, case studies, editorials, opinion papers, reviews, protocols, modelling studies, and systematic or narrative reviews
Type of paper	English language	Grey literature
	Peer-reviewed, published literature	
	Published from January 2000–July 2020	

Diurnal temperature range—difference between daily minimum and maximum temperatures.

**Table 2 ijerph-19-09109-t002:** Summary of study findings that assess temperature effects on all-cause infant mortality and SIDS.

Main Author (Year)	Research Design	Location, Year of Study	Description of Climate	Main Temperature Exposure Variable(s)	Statistical Analysis	Outcomes Measure (Source of Data)	Effect Estimates
All-Cause Mortality							
Basu (2008) [35]	Case-crossover	California, USA 1999–2003	Mean apparent temperature in warm season: 21.4 °C	Mean daily apparent temperature. Lag 0 days	Time-stratified case-crossover	Nonaccidental infant death. Sample size not provided for this age group (routine data)	4.9% (95%CI = 1.0, 7.1%) increase in mortality for every 4.7 °C increase in temperature (no threshold presented).
Basu (2015) [21]	Case-crossover	California, USA, 1999–2011	Mean temperature in warm season in coastal area, 17.8 °C and in non-coastal area 20.8 °C	Mean daily apparent temperature in warm season. Lag 0–3 days	Time-stratified case-crossover	12,356 infant deaths (routine data)	Increase in all-cause mortality by 4.4% (95%CI = −0.3, 9.2) per 5.6 °C increase temperature (no threshold presented). For stratified analysis; all-cause mortality in black race/ethnic group: 13.3% (95%CI = 0.6, 27.6).
Schinasi (2020) [15]	Case-crossover	Philadelphia, USA, 2000–2015	Humid, subtropical climate. Mean dry bulb temperature 23.3 °C in warm season	Minimum daily temperature in warm season. Lag 0–3 days	Time-stratified case-crossover	1522 all-cause infant deaths (routine data)	OR comparing 23.9 °C and 26.1 °C with 4.4 °C were 2.1 (95%CI = 1.2, 3.6) and 2.6 (95%CI = 1.3, 5.0), respectively. Risk of infant mortality increased by 22.4% (95%CI = 5.0, 42.6) for every 1 °C increase in temperature above 23.9 °C. No evidence of effect modification.
Son (2017) [36]	Retrospective cohort	7 cities in South Korea, 2004–2007	Climate varies across cities. Average 13.0 °C–14.8 °C.	Mean daily temperature averaged 2 weeks before death	Cox proportional hazards model	557 all-cause infant mortality and SIDS (routine data)	HR for 1 °C increase in temperature during 2 weeks before death; 1.51 (95%CI = 1.45, 1.56) for total mortality and 1.50 (95%CI = 1.35, 1.66) for SIDS. No evidence of effect modification.
Basagaña (2011) [37]	Case-crossover	Catalonia, Spain, 1983–2006	Mediterranean climate. Average maximum temperature 19.9 °C–27.4 °C. 95th percentile of maximum temperature in warm season was 27.3 °C–38.0 °C	Maximum daily temperature during warm season over 95th percentile. Lag 0–6 days	Time-stratified case-crossover	3144 deaths on hot days (routine data)	RR 1.25 (95%CI = 1.02, 1.53) on hot day as compared to non-hot day.
Fouillet (2006) [34]	Heat episode analysis	France, August 2003	Temperate oceanic climate. During heatwave, temperature exceeded 35° for at least 9 days in most French departments	Daily minimum and maximum temperatures for very hot days	Excess mortality (observed over expected deaths)	Excess mortality in infant population (0.4 million) (routine data)	Excess of 25 deaths in male infants; mortality ratio 1.3 (95%CI = 1.0, 1.6) of observed over expected deaths during same time period. No excess mortality observed in females.
Diaz (2004) [38]	Retrospective cohort	Madrid, Spain, 1986–1997	Mediterranean climate. Mean maximum temperature 19.7 °C.	Maximum, minimum, and temperature in cold wave. Lag 0–7 days	Poisson regression models	Infant mortality (sample size not provided for this age group) (routine data)	17.4% of deaths are attributable to cold wave. RR 1.21 (95%CI = 1.10, 1.32) per 1 °C, below 6 °C, at lag 4 days.
Karlsson (2020) [32]	Retrospective cohort	Northern Sweden 1860 –1899	Sub-artic climate. Mean monthly temperatures 14.8 °C–15 °C	Mean daily temperature. No lags.	Time-event binomial regression model	330 neonatal deaths (parish registers)	Temperature at and after birth not associated with increased risk of neonatal mortality for whole study population. Sami infants had a higher mortality risk than winter-born non-Sami infants with lower temperatures on day of birth, HR 1.46 (95%CI = 1.07, 2.01).
Scalone (2018) [33]	Retrospective cohort	Northern Italy, 1820–1900	Prolonged rainy and dry periods with frequent violent weather events. Mean temperatures 1.2 °C–20.7 °C	Minimum daily temperature. No lag.	Multivariate statistical analysis, using event-history techniques	175 neonatal deaths (parish registers)	Temperature at birth had a significant effect on neonatal mortality; RR 0.911 SE 0.028, *p* = 0.003; 9% decrease in mortality for each unit increment in temperature at birth. Daily temperature was not associated with increased neonatal mortality RR 1.031 SE 0.928 *p* = 0.255. Landless rural labourers were at a higher risk for neonatal mortality related to temperature at birth as compared to sharecroppers and farmers during the 1860–1900 period: RR 0.830 SE 0.068 *p* = 0.023
SIDS							
Jhun (2017) [39]	Case-crossover	210 cities in USA 1972–2006	8 climate clusters were created given variability across different cities	Mean temperature. Lag 0–2 days	Time-stratified case-crossover	60,364 SIDS cases (routine data. ICD 8–795.0, ICD 9–798.0 and ICD 10-R95.0)	8.6% (95%CI = 3.6, 13.8) increase in SIDS risk for 5.6 °C increase in temperature. 3.1% (95% CI = −5.0, −1.3) decrease in the winter. Summer risks were greater among black infants 18.5% (95%CI = 9.3, 28.5) compared to white infants 3.6% (95%CI = −2.3, 9.9)
Auger (2015) [40]	Case-crossover	Montreal, Canada1981–2010	Continental climate with hot summers and cold winters. Maximum temperatures ranged from −1.5 °C to 33.8 °C on days before SIDS occurred.	Maximum temperature during warm months. Lag 0–2 days	Time-stratified case-crossover	196 SIDS cases (routine data)	Same-day maximum daily temperature of 24 °C, 27 °C and 30 °C when compared to 20 °C increased the odds of SIDS by 1.41 (95%CI = 1.17, 1.69), 2.12 (95%CI = 1.43, 3.14) and 3.18 (95%CI = 1.76, 5.77), respectively.
Waldhoer (2017) [41]	Case-crossover	Vienna, Austria1984–2014	Continental climate. Mean maximum temperatures on day before and day of SIDS 14.4 °C–28.9 °C	Maximum daily temperature. Lag 0–1 days	Time-stratified case-crossover	187 SIDS cases (routine data. ICD 10-R95.0, ICD 9–798.0 and mentioned autopsy)	OR for SIDS at same day temperatures of 24 °C, 27 °C and 30 °C compared to 20 °C were 1.05 (95%CI = 0.87, 1.27), 1.13 (95%CI = 0.76, 1.68), and 1.23 (95%CI = 0.67, 2.29), respectively.
Scheers-Masters (2004) [42]	Retrospective cohort	Arkansas, Georgia, Kansas and Missouri, USA1980	Temperate climate	Daily average and maximum temperatures	Chi-squared test for trend. Spearman’s rank correlation coefficient	111 SIDS cases (ICD 9–79.8) and heat-related mortality (ICD 9–900: Death due to excessive heat due to weather conditions)	No increase in SIDS rate with increasing average (*p* = 0.713) and maximum temperature (*p* = 0.362).
Chang (2013) [43]	Case-control	Taiwan 1994–2003	Temperate climate	Daily maximum temperature categorised into percentiles	Log-liner model	1671 SIDS cases (routine data, ICD 9–789)	Risk of SIDS at the lowest percentile <5th (9.2–14.2 °C) compared to 45th–55th percentile (21.9–23.3 °C) is 2.10 (95%CI = 1.67, 2.64). Daily mean temperature in 85th–95th percentile (26.4–27.3 °C) and >95th percentile (27.3–33.2 °C) associated with reduced risk of SIDS 0.60 (95%CI = 0.46, 0.79) and 0.61 (95%CI = 0.50, 0.75), respectively.

CI–confidence interval; OR—odds ratio; HR—hazard ratio; RR—relative risk; SE—standard error; SIDS—sudden infant death syndrome; lag—effects of temperature may reflect exposure on the preceding days. 
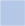
 Cold exposure.

**Table 3 ijerph-19-09109-t003:** Summary of the study findings that assess temperature effects on hospital visits/admissions, infectious diseases and other neonatal outcomes.

Main Author (Year)	Research Design	Location, Year of Study	Description of Climate	Main Temperature Exposure Variable(s)	Statistical Analysis	Outcomes Measures (Source of Data)	Effect Estimates
All-Cause/Heat-Related Hospital Visits and/or Admissions							
Sheffield (2018) [13]	Case-crossover	New York City, USA 2005–2011	Continental climate.	Minimum, maximum and average apparent daily temperature. Lag 0–6 days	Time-stratified case-crossover	278,114 all-cause emergency department (ED) visits (hospital records, all-cause diagnostic codes)	Positive association with maximum temperature; 0.6% (95%CI = 0.1, 1.1).
Xu (2017) [44]	Time-series analysis	Brisbane, Australia2005–2015	Humid, sub-tropical climate. Mean temperature 9.0 °C–30.9 °C	Mean daily temperature. Nine different heatwave definitions	Poisson generalised additive model and distributed lag non-linear model	53,792 all-cause hospital admissions (hospital records)	When mean temperature was >97th percentile, the RR of hospital admissions increased by 1.05 (95%CI = 1.01, 1.10), and then further increased to 1.18 (95%CI = 1.05, 1.32) when duration of heatwave increased from 2 days to 4 days. No evidence of effect modification.
Kakkad (2014) [19]	Retrospective cohort	Ahmedabad, India 2009–2011	Warm, dry conditions that often include heat waves. Average monthly maximum of 38.8 °C between March and June. Heat wave in May 2010, maximum temperature 46.8 °C.	Daily maximum temperature. No lags.	Generalised linear models and segmented regression.	Heat-related illness in neonatal admissions—defined as diagnosis of exclusion when body temperature >38 °C with any signs and symptoms such as refusal to feed, signs of dehydration, increased respiratory rate, convulsions and/or lethargy (hospital records)	Above 42 °C, each temperature increase of a degree was associated with a 43% increase in heat-related admissions (95%CI = 9.2, 88).
Mannan (2011) [28]	Retrospective cohort	Sylhet district, Bangladesh2004–2006	Tropical monsoon climate. Monthly average temperatures range 19.3 °C–29.3 °C for Sylhet and 17.7 °C–29.5 °C for Mirzapur	7 day rolling average temperature and rolling humidity index prior to diagnosis	Multivariable logistic regression.	Very severe disease in 6936 newborns in Sylhet and 5900 newborns in Mirzapur—diagnosed on history using clinical algorithms) (data from 2 cluster randomised controlled trials)	OR for temperature at time of diagnosis of very severe disease were 1.14 (95%CI = 1.08, 1.21) in Sylhet and 1.06 (95%CI = 1.04, 1.07) in Mirzapur.
Infectious diseases							
Gosai (2009) [45]	Retrospective cohort	Auckland, New Zealand1994–2004	Temperate oceanic climate. Average monthly temperatures 10.8 °C–19.8 °C	Daily minimum temperature. Lag 0–7 days	Pearson’s correlation	Hospital admissions for respiratory illness (hospital admissions data. No ICD codes given)	Correlation between minimum temperature and respiratory infections and inflammation (r = −0.42 and *p* < 0.001) and total whooping cough and acute bronchiolitis (r = −0.40 and *p* < 0.001).
Yan (2019) [29]	Time-series analysis	Shenzhen, China 2009–2017	Subtropical monsoon climate. Mean temperature over period 23.3 °C.	Daily temperature, unclear which index used. Lag with a max of 30 days	Quasi-Poisson regression based on distributed lag nonlinear model	50,657 HFMD cases (surveillance data—notifiable disease)	Cumulative RR of HFMD over 14 days in 0–1 age group was RR 0.58 (95%CI = 0.4, 0.84) for temperature in 5th percentile and RR 2.03 (95%CI = 1.77, 2.33) in the 95th percentile. 5th percentile was 12.9 °C, 95th was 30 °C and median 24.6 °C.
Yin (2015) [30]	Time series analysis	Chengdu, China 2013–2014	Humid sub-tropical climate. Mean temperature for study period 16.21 °C.	Daily mean temperature. Lag 0–14 days	Poisson generalised linear regression combined with distributed lag non-linear model	74,247 HFMD cases aged 0–5 years (surveillance data—notifiable disease)	Risk of HFMD significantly increased at temperatures 14, 17.2, 23.2 and 27 compared to 0 °C at lag 0 and lag 14 for infants <1 years old. No increase in HFMD risk for colder temperature exposure of 4.1 °C. For lag 0, RR 1.15 (95%CI = 1.03, 1.29), 1.19 (95%CI = 1.06, 1.34), 1.29 (95%CI = 1.13, 1.44), and 1.31 (95%CI 1.16, 1.48) for 14, 17.2, 23.2, and 27 °C, respectively. For lag 14 RR 1.09 (95%CI = 1.03, 1.16), 1.07 (95%CI = 1.00, 1.14), 1.06 (95%CI = 1.00, 1.13), and 1.03 (95%CI = 1.02, 1.04) for 14, 17.2, 23.2, and 27 °C, respectively.
Nenna (2017) [46]	Prospective cohort	Rome, Italy 2004–2014	Mediterranean climate	Weekly average temperature. No lag	Pearson’s correlation	723 cases of viral bronchiolitis in hospitalised infants (prospective clinical records)	The number of RSV-positive infants correlated negatively with temperature (r = −0.46, *p* < 0.001).
Kim (2017) [47]	Prospective cohort	Cheonan, South Korea 2006–2014	Subtropical climate	Mean daily temperature. No lag days	Logistic regression	2484 infants admitted with RSV A and RSV B (lab confirmed admissions with respiratory symptoms)	RSV A and RSV B infections were negatively correlated with average temperature; −0.056 for RSV A and −0.069 for RSV B infection, with *p* < 0.001 for both.
Hoeppner (2017) [48]	Retrospective cohort	Australia and New Zealand 2009–2011	Perth—Mediterranean, Melbourne and Auckland-oceanic and Brisbane—humid, subtropical	Minimum temperature aggregated over a week. Lag 0–4 weeks	Linear regression. Poisson and negative binomial regression to verify results	3876 infants admitted with bronchiolitis (data from prospective, multicentre clinical trial)	Minimum temperature and lag 0): r −0.62 (−0.75, 0.48) *p* < 0.001. Lag 1: r −0.58 (95%CI = −0.75, −0.43) *p* < 0.001. Lag 2: r −0.67 (95%CI = −0.75, −0.58) *p* < 0.001. Lag 3: r −0.34 (95%CI = −0.49, −0.20) *p* < 0.001
Other neonatal outcomes							
Iijima (2016) [49]	Prospective cohort	Hamamatsu, Japan 2012–2013	Temperate climate. Average temperatures spring 15.3 °C, summer 27.3 °C, autumn, 18.0 °C and winter 7.0 °C.	Mean outdoor and indoor, and wind chill temperature. No lag	Simple and multivariate regression	498 neonates. International normalised ratio on day 4 after birth (data from healthy neonates)	Significant correlation between INR and outdoor temperature (r = 0.25, *p* < 0.001). Weakly negative correlation between INR and room temperature (r = −0.13, *p* = 0.02).
Scrafford (2013) [31]	Retrospective cohort	Southern Nepal 2003–2006	Humid subtropical climate	Minimum daily temperature. No lag days	Bivariate and multivariate analyses	Incidental jaundice in 18,985 neonates, defined as first report of yellow eyes/body based on visual assessment by study staff, not laboratory confirmed (part of nested pair of cluster-randomised, placebo-controlled, community based clinical trial)	OR 1.03 (95%CI = 1.02, 1.05) *p* < 0.001 for each 1 °C increase in minimum ambient temperature. Adjusted OR 1.04 (95%CI = 1.03, 1.06).

RSV—Respiratory Syncytial Virus; HFMD—Hand, Foot, and Mouth Disease; CI—confidence interval; OR—odds ratio; RR—relative risk; INR—International Normalised Ratio. 
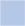
 Cold exposure.

## Data Availability

No new data were created or analyzed in this study. Data sharing is not applicable to this article.

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
