# Peer review of "The Effect of High and Low Ambient Temperature on Infant Health: A Systematic Review"

_ijerph, 2022, doi:10.3390/ijerph19159109_

Round 1

Reviewer 1 Report

Well conducted review and very nice topic.

Some matters of concern:

- Introduction: please discuss more on the effect of socioeconomic status and, as it is pivotal when assessing the overall impact of high or low temperatures on infant health, hence by enhancing or lowering the risk. How many papers did explore this topic in your review?

- Methods: I think it may be useful to divide between metropolitan areas or not, as access to hospital,  green areas and so on may be quite different. How did you suppose to compare studies with contradictory results?

- Results: how do you comment on the apparent absence of a worsening trend of any of the assessed topic among years despite the increasing burden in the climate change?

Author Response

Dear Reviewer 

Thank you for giving us the opportunity to submit a revised draft of our manuscript. We appreciate the time and effort that you have dedicated to providing your valuable feedback. We have been able to incorporate changes to reflect most of the suggestions provided by the reviewers and have highlighted those changes in the manuscript. Here is a point-by-point response to your comments and concerns.

Comment: Introduction: please discuss more on the effect of socioeconomic status and, as it is pivotal when assessing the overall impact of high or low temperatures on infant health, hence by enhancing or lowering the risk. How many papers did explore this topic in your review?

Response: We have included a short discussion on socioeconomic status in the introduction, lines 44-51. Five studies assessed the impact of socioeconomic factors on temperature and infant health which found conflicting evidence. This is discussed in the results, lines 211-223 and key findings, lines 284-300.

Lines 44-51 in the introduction that discuss socioeconomic status in temperature-health associations: “Additionally, low socioeconomic and ethnic minority groups are at a higher risk of temperature-related adverse health outcomes [3,5–7,56]. Socioeconomic status may impact on factors such as access to air-conditioning and to healthcare, which may differ across countries, regions and urban versus rural populations [3,5,7,56]. Moreover, populations from a lower socioeconomic backgrounds, and urban areas are more likely to reside in warmer neighbourhoods due to high population density, sparse vegetation, and lack of green spaces, and are less likely to be able to adapt to temperature extremes [3].”

Comment: Methods: I think it may be useful to divide between metropolitan areas or not, as access to hospital, green areas and so on may be quite different. How did you suppose to compare studies with contradictory results?

Response: Thanks for this suggestion. The effect modifiers such as urban areas can be impacted by other factors such as differences across countries (like LMIC’s and HIC’s) and individual factors such as socioeconomic factors and race. For this reason, we don’t think we can divide studies on this risk factor in the analysis.

We have however made edits to results (lines 211-223) and discussion (lines 285-300) to ensure that this topic is discussed from evidence collected in our review.  

Comment: Results: how do you comment on the apparent absence of a worsening trend of any of the assessed topic among years despite the increasing burden in the climate change?

Response: This is a neglected topic in the literature, and unfortunately, there is insufficient evidence in our literature review to comment on any differences, for the following reasons. Firstly, we excluded studies that were published before 2000, which limits our ability to comment on longer time trends. Secondly, in reviewing column 3 in Tables 2 and 3, most studies have a wide range of data from 1990’s to 2000’s, and do not stratify by time periods. Thirdly, infant and child mortality has improved in recent years, and adaptative measures could have reduced the impacts of temperature on infant health, especially for cold. For this reason, we might expect the burden to decrease, rather than increase. I have added a sentence in the results about the lack of evidence in differences over time in lines 222-223: “There is no evidence of a difference in magnitude of temperature-infant health outcomes over time, but this is an important issue to consider.”

Reviewer 2 Report

Lakhoo et.al conducted a systematic review to explore the association between acute adverse infant outcomes (children 0-1 years) and exposure to high and low ambient temperatures. They concluded that higher temperatures were associated with increased risk of acute infant mortality, hospital admissions, and hand, foot, and mouth disease, while low temperature impacts on infant mortality and episodes of respiratory disease. The associations between temperature and sudden infant death syndrome were inconsistent. Here I have several questions and suggestions for the authors to consider.

Major:

1.     Lines 51 to 65: The authors described why infants (children aged 0-1) are more vulnerable to temperature extremes. It should be noted that thermoregulation of children aged 0-1 was mainly affected by their caregiver. I wonder if it is reasonable to investigate the association between temperature and infant health among children aged 0-1? Please discuss.

2.     The Figure 2 is confusing. For example, after removing duplicate records, a total of 2396 studies were identified; however, the number of studies after being screened were still 2396. Please clarify.

3.     In Tables 2 and 3, the authors listed all included studies. I would suggest that they simplify their words appropriately and further add more information for the statistical analysis and lag period. In addition, I noted that a majority of included studies had conducted quantitative dose-response analysis, and maybe a mate-analysis can be performed to provide more quantitative information for the association of temperature with infant health.

4.     In Table 4, the author provided a summary of the critical appraisal using a modified OHAT Risk of bias tool. More information such as the specific definition for different risk levels need be provided in main text. 

Minor:

1.     Please unify the usage of numbers in the full manuscript, such as lines 114 and 116: “Five” and “5”.

Please simplify the expression in the main text, especially in the “Key findings” and “Conclusions” sections. I suggest that the authors carefully revise the manuscript before its publication.

Author Response

Dear Reviewer

Thank you for giving us the opportunity to submit a revised draft of our manuscript. We appreciate the time and effort that you have dedicated to providing your valuable feedback. We have been able to incorporate changes to reflect most of the suggestions provided by the reviewers and have highlighted those changes in the manuscript. Here is a point-by-point response to your comments and concerns.

Comment: Lines 51 to 65: The authors described why infants (children aged 0-1) are more vulnerable to temperature extremes. It should be noted that thermoregulation of children aged 0-1 was mainly affected by their caregiver. I wonder if it is reasonable to investigate the association between temperature and infant health among children aged 0-1? Please discuss.

Response: In lines 71 -76, we address the importance of the caregiver role in the mechanisms for temperature effects: “In addition to these physiological and anatomical factors, infants are particularly vulnerable, as they rely on a caregiver to safeguard them, which includes  moving to a cooler or warmer environment if required, dressing appropriately, and for keeping hydrated [14,18]. We address the potential mechanisms for temperature-health associations.

Comment: The Figure 2 is confusing. For example, after removing duplicate records, a total of 2396 studies were identified; however, the number of studies after being screened were still 2396. Please clarify.

Response: Thank you for your comment. We have reviewed Figure 2 and removed the unnecessary box and added the number of records that were excluded at the screening stage, which has been cut off. (Figure 2, page 7)

Comment: In Tables 2 and 3, the authors listed all included studies. I would suggest that they simplify their words appropriately and further add more information for the statistical analysis and lag period. In addition, I noted that a majority of included studies had conducted quantitative dose-response analysis, and maybe a mate-analysis can be performed to provide more quantitative information for the association of temperature with infant health.

Response: Thank you for your suggestions. We have added in additional columns in the table that address statistical analysis methods and lag periods assessed. The column headings are main temperature exposure variable(s) and statistical analysis. In addition, we have reviewed the table and simplified wording where possible.

The authors do see the value in being able to summarise the data using a meta-analysis, but do not think this is feasible with this review. There is significant methodological diversity in the studies that have been presented in the paper. There is diversity in outcomes measures, study designs, statistical analysis and effect estimates. For example, if we explore a meta-analysis on mortality, there are significant differences in how the temperature-mortality association is measured. One studies measures risk with every 4.7°C increment increase, another every 5.6°C increase, and lastly, another study with 1°C degree increase in temperature. There are also different thresholds that are used, for example, Diaz et al, measures mortality per 1°C decrease in temperature, below 6°C. Combining these studies may obscure the genuine similarities or differences due to the methodological diversity and statistical heterogeneity. We have added a sentence in methodology, lines 129-130 to explain why a meta-analysis was not possible: “The considerable methodological diversity and statistical heterogeneity of study findings precluded meta-analysis.”

Comment: Table 4, the author provided a summary of the critical appraisal using a modified OHAT Risk of bias tool. More information such as the specific definition for different risk levels need be provided in main text. 

Response: More detailed information on the OHAT risk of bias scoring process has been added into the main text in lines 111-125: “An adapted quality assessment tool was developed to assess the quality of the studies using the Office of Health Assessment and Translation (OHAT) tool [23], which considers the risk of bias. Two further questions on analysis and conflicts of interest were added to the tool based on a systematic review that recommended the inclusion of nine domains for observational researchselection, exposure, outcome assessment, confounding, loss to follow-up, analysis, selective reporting, conflicts of interest and other [24]. OHAT does not specifically consider environmental health study designs like time series and case-crossover, hence we adapted the tool to tailor it to our research question (See Table S2) Each domains was scored as a) definitely low b) probably low c) probably high or insufficient information and d) definitely high [23]. The final grading was given dependent on the individual domain grades. We considered an individual study to have a definitely low risk or probably low risk of bias if all, or most domains were rated as definitely low risk, respectively. Similarly, probably high risk and definitely high risk were rated based on the number of domains that scored probably or definitely high risk.”

Comment: .     Please unify the usage of numbers in the full manuscript, such as lines 114 and 116: “Five” and “5”.

Response: Thanks, we have reviewed the manuscript and corrected the usage of numbers.

Comment: Please simplify the expression in the main text, especially in the “Key findings” and “Conclusions” sections. I suggest that the authors carefully revise the manuscript before its publication.

Response: Many thanks for this suggestion. We have made a large number of edits throughout the manuscript to simplify the text and improve clarity

Reviewer 3 Report

There are a few comments / suggestions:

Figure 2: Please include an n number for studies excluded based on title and abstract (the number of studies).

I like the data presented in table 2. Would it be possible to present the data in an additional graphic presentation? I would suggest to provide a graph with temperature increase or temperature decrease on the x-axis and the increase in infant mortality on the y-axis. Thus a correlation as suggested by the individual quotations with higher and lower temperatures resulting in an increased overall mortality could be demonstrated visually. The size of the individual studies could be expressed by a bubble size in this graph. Alternatively a forest plot summarizing data in table 2 could be used to visualize the data.

In the discussion I would suggest to add a point discussing the potential magnitude of the effect of ambient temperature changes on infant mortality rates (1. By comparing differences in mortality rates in high income countries and low income countries and 2. By comparing historic changes in SIDS incidence [e. g. Obladen M. Cot Death: History of an Iatrogenic Disaster. Neonatology. 2018;113(2):162-169. doi: 10.1159/000481880. Epub 2017 Dec 15. PMID: 29241201.] relative to the temperature related changes in SIDS incidence as assumed by interpretation of the literature data in this review).

Author Response

Dear Reviewer 

Thank you for giving us the opportunity to submit a revised draft of our manuscript. We appreciate the time and effort that you have dedicated to providing your valuable feedback. We have been able to incorporate changes to reflect most of the suggestions provided by the reviewers and have highlighted those changes in the manuscript. Here is a point-by-point response to your comments and concerns.

Comment: Figure 2: Please include an n number for studies excluded based on title and abstract (the number of studies).

Response: Thanks so much for pointing this out. Apologies, the number got cut off when the diagram was converted into a figure. We have amended Figure 2, by removing unnecessary boxes and adding the number of records that were excluded at the screening stage. (Figure 2, page 7)

Comment: I like the data presented in table 2. Would it be possible to present the data in an additional graphic presentation? I would suggest to provide a graph with temperature increase or temperature decrease on the x-axis and the increase in infant mortality on the y-axis. Thus a correlation as suggested by the individual quotations with higher and lower temperatures resulting in an increased overall mortality could be demonstrated visually. The size of the individual studies could be expressed by a bubble size in this graph. Alternatively a forest plot summarizing data in table 2 could be used to visualize the data.

Response: The authors do see the value in being able to summarise the data using a meta-analysis, or forest plot figures, but do not think this is feasible with this review. There is significant methodological diversity in the studies that have been presented in the paper. There is diversity in outcomes measures, study designs, statistical analysis and effect estimates. If these diverse studies are combined, it may not yield a meaningful summary estimate of effect. For example, if we explore a meta-analysis on mortality, there are significant differences in how the temperature-mortality association is measured. One studies measures risk with every 4.7°C increment increase, another every 5.6°C increase, and lastly, another study with 1°C degree increase in temperature. There are also different thresholds that are used, for example, Diaz et al, measures mortality per 1°C decrease in temperature, below 6°C. Combining these studies may be meaningless and obscure the genuine similarities or differences due to the methodological diversity. Similarly, the same methodological diversity will affect bubble plots and line graphs to estimate overall effect sizes.

Comment: In the discussion I would suggest to add a point discussing the potential magnitude of the effect of ambient temperature changes on infant mortality rates (1. By comparing differences in mortality rates in high income countries and low income countries and 2. By comparing historic changes in SIDS incidence [e. g. Obladen M. Cot Death: History of an Iatrogenic Disaster. Neonatology. 2018;113(2):162-169. doi: 10.1159/000481880. Epub 2017 Dec 15. PMID: 29241201.] relative to the temperature related changes in SIDS incidence as assumed by interpretation of the literature data in this review).

Response:

Thank you for the comment and reference shared. I think in your comment about magnitude of effect of temperature on infant mortality, you are referring to attributable risk, which is only measured in one of the 21 studies included in this review. Instead, most studies measure the odds or the risk of adverse events if the temperature is high or low, but not the contribution of temperature to overall infant mortality rates. We would not be able to comment on that using the data in this review.

If you are referring to effect modification, we are limited in making conclusions on the differences between LMIC’s and HIC’s due to a limited number of studies in LMIC’s, and the heterogeneity in outcomes and how they are measured. However, in lines 302-312, in the discussion, we have edited some text to discuss the difference in results between LMIC’s and HIC’s: “Only five of the 21 included studies were from LMICs and entire regions such as Africa, the Middle East and South and Central America are not represented. All five studies found a positive association between temperature and adverse infant outcomes, however, comparison of the magnitude of difference to high-income countries is limited due to the spread of LMIC studies across different outcomes and methodological heterogeneity. However, there is evidence, outside of this review, that the impact of climate change on children is not evenly distributed, but occurs in LMICs that are already experiencing a higher burden of disease [19]. There is an obvious mismatch between vulnerability to temperature effects and research effort, which needs to be addressed in future work.”

Unfortunately, this study has a limitation in the review of SIDS. We did not include studies published before 2000, resulting in missed literature on SIDS, especially prior to the change in advice from prone sleeping. Further, no studies in this review compared the incidence before and after advice was changed, even though prone sleeping is one of the hypothesized reasons for a heat-SIDS association. The limitations of this study in assessing SIDS is highlighted in the limitations section, lines 324-328.  

Round 2

Reviewer 2 Report

The authors have addressed most of my concerns, except they stated that they were unable to conduct meta-analysis due to the difference of exposure increments. This is not a problem at all. "One studies measures risk with every 4.7°C increment increase, another every 5.6°C increase, and lastly, another study with 1°C degree increase in temperature". The risk estimates with different exposure increments can be easily re-calculated for a given exposure increment.